# RNA-Independent Regulatory Functions of lncRNA in Complex Disease

**DOI:** 10.3390/cancers16152728

**Published:** 2024-07-31

**Authors:** Michaela Kafida, Maria Karela, Antonis Giakountis

**Affiliations:** Department of Biochemistry and Biotechnology, University of Thessaly, Biopolis, Mezourlo, 41500 Larissa, Greece

**Keywords:** lncRNA DNA loci, cancer, CRISPR-i, PAS, chromatin architecture, chromatin loops, epigenetics, transcriptional regulation

## Abstract

**Simple Summary:**

Recent studies highlight the complexity of long non-coding RNA (lncRNA) genetic loci in the pathogenesis of complex diseases, including carcinogenesis and cancer progression. These studies converge on the existence of RNA-independent functional layers that participate in the underlying regulatory mechanism of the corresponding lncRNAs. Such layers include downstream regulation through embedded DNA elements, nascent transcription of the lncRNA locus, or even their combined function. Consequently, it is insufficient to limit our investigation to the potential function of a lncRNA transcript if we want to fully uncover the regulatory role of its locus.

**Abstract:**

During the metagenomics era, high-throughput sequencing efforts both in mice and humans indicate that non-coding RNAs (ncRNAs) constitute a significant fraction of the transcribed genome. During the past decades, the regulatory role of these non-coding transcripts along with their interactions with other molecules have been extensively characterized. However, the study of long non-coding RNAs (lncRNAs), an ncRNA regulatory class with transcript lengths that exceed 200 nucleotides, revealed that certain non-coding transcripts are transcriptional “by-products”, while their loci exert their downstream regulatory functions through RNA-independent mechanisms. Such mechanisms include, but are not limited to, chromatin interactions and complex promoter-enhancer competition schemes that involve the underlying ncRNA locus with or without its nascent transcription, mediating significant or even exclusive roles in the regulation of downstream target genes in mammals. Interestingly, such RNA-independent mechanisms often drive pathological manifestations, including oncogenesis. In this review, we summarize selective examples of lncRNAs that regulate target genes independently of their produced transcripts.

## 1. Introduction

For a long time, messenger RNAs (mRNAs) have been at the very center of basic research focus, given their key role in cellular physiology. Yet, metagenomic era transcriptome analysis uncovered a variety of non-coding RNAs (ncRNAs) that collectively encapsulate the diversity of molecular operation of the higher eukaryotes. However, the existence of these non-coding transcripts is not new. In fact, ncRNAs were first discovered back in the 50s and were referred to as the housekeeping ncRNAs (today known as transfer- and ribosomal- RNAs, or tRNAs and rRNAs, respectively). However, it was not until the ’00s that the totality and wealth of the ncRNA class were assessed with the development of next-generation sequencing (NGS) platforms. At first, ncRNAs were considered as transcriptional by-products and were quickly neglected as non-functional “junk DNA”, a premature belief that was quickly overturned with the development of more advanced sequencing technologies. In fact, global research consortia including the Human Genome Project (HGP, [1]) and the Encyclopedia of DNA Elements (ENCODE, [2]) led to the detection of a series of ncRNAs, overall supporting that most of the mammalian DNA can be transcribed into ncRNAs that significantly outnumber coding genes.

Today, ncRNAs are classified into two major categories, housekeeping and regulatory ncRNAs, depending on the roles they exert [3]. Housekeeping ncRNAs directly participate in the transcription and translation of mRNAs and primarily constitute tRNAs, rRNAs, snRNAs (small nuclear RNAs mediating splicing [4]), and snoRNAs (small nucleolus RNAs that are responsible for RNA modifications [5]). On the other hand, regulatory ncRNAs that govern transcription and translation of target genes are predominantly composed of miRNAs (microRNAs that hybridize to a target mRNA preventing its translation or promoting its degradation [6]), circRNAs (circular RNAs known to serve as transcription regulators, miRNA decoys and scaffolds [7]), piRNAs (PIWI-interacting RNAs that regulate gene expression both in somatic and germ cells [8]), and lncRNAs (long non-coding RNAs with diverse roles in development and pathologies [9]). All these ncRNAs are cataloged in detail in public databases such as the NONCODEV5 [10], Rfam [11], circBase [12], and miRbase [13].

Herein, we focus on lncRNAs as they constitute 81.8% of all regulatory ncRNAs that have been identified [14]. LncRNAs refer to mature transcripts that are longer than 200 nucleotides with low or no protein-coding potential, although some lncRNAs have been shown to embed ORFs (open reading frames) in their sequence, encoding for unusually small yet functional peptides [15]. Their biogenesis relies on the combined function of RNA polymerase II (RNA pol II), along with standard post-transcriptional processing of the nascent transcript in the form of 5′ m7G cap, 3′ polyA tail addition, and intron splicing [16]. However, unlike their coding counterparts, not all mature lncRNA transcripts are exported to the cytosol. Nuclear-retained lncRNAs persist in the nucleoplasm in which they frequently interact either with nuclear regulatory protein partners or with the chromatin itself [17]. In terms of locus classification, lncRNAs can be globally divided into intergenic if their locus does not span other loci and intragenic when their locus overlaps with other loci. The latter can be further subdivided into intronic and exonic transcripts depending on the location of the lncRNA locus inside the host gene, while intragenic lncRNAs can be either sense or antisense depending on the orientation of their transcription compared to the host locus. Finally, lncRNAs that originate from unspliced loci are referred to as macro lncRNAs, whereas when their sequence retains introns of coding genes, they are classified as retained intron lncRNAs [18].

LncRNAs play fundamental roles in a series of cellular processes, both physiological and pathological. For instance, H19 and Xist, which were the first lncRNAs to be discovered, are now notorious for their crucial epigenetic modulation [19,20]. Xist is responsible for X chromosome inactivation in mammals, since it accumulates over one copy of the X chromosome, preventing most of its transcriptional activity [21]. Following the discovery of H19 and Xist in 1990, tens of thousands of lncRNAs have been detected and their numbers are still growing. Some of them are involved in development, cell differentiation, and generally in normal physiology [22,23], whereas others have been studied as part of several pathologies [24,25]. Amongst others, it is worth mentioning that a plethora of lncRNA transcripts have been found to actively participate in tumorigenic processes [26,27].

Importantly, recent studies indicate that lncRNA-mediated regulation of downstream targets is far more complex, encapsulating embedded regulatory elements within the ncDNA locus that drive independent functional roles, which can oppose those mediated by the mature ncRNA transcript itself. In fact, the non-coding transcript has been frequently described as the “by-product” of an RNA-independent mechanism of ncRNA function. Below we summarize selective studies, organized according to the level of involvement of the corresponding transcript in the regulation of pathological or physiological processes, which collectively highlight the role of ncRNA-mediated function beyond the level of RNA. We took particular care in describing in detail the functional experiments that support the multi-faceted ncRNA-mediated regulation, hoping that this will serve as a guide that can fuel the discovery of additional examples of complex ncRNA function.

To ensure the delivery of a comprehensive review of the identified lncRNAs that exert RNA-independent actions, our search lasted over four months (1 February 2024 to 31 May 2024). The bibliographic search was conducted in PubMed and Google Scholar using a combination of keywords such as “lncRNA”, “ncRNA”, “cancer”, “disease”, “locus”, “locus function”, “DNA elements”, “nascent transcription”, “epigenetics”, and “chromatin conformation”. To avoid language restrictions, only English publications were included. For the first screening of the literature, we focused on titles and abstracts of original research articles and systematic reviews, and subsequently, the complete text was examined to ensure compatibility with the subject. Finally, the list of our selected citations was thoroughly reviewed to further uncover relevant original research articles and meta-analyses, which we evaluated again as described above.

## 2. The Multifaceted Regulatory Mechanism of lncRNAs

Several studies focus on the identification and characterization of lncRNAs, which are associated with a variety of diseases. For example, a growing number of lncRNAs has been linked to the deadly pancreatic cancer [28] or other forms of cancer [26], whereas other lncRNAs that are implemented are unrelated to cancer diseases, such as rheumatoid arthritis [29], which associate with the immune system. In most cases, the vast majority of the identified lncRNA loci produce transcripts that exert downstream regulatory functions. Such an example is the lncRNA NALT1 which is correlated with acute lymphoblastic leukemia [30]. This non-coding transcript targets NOTCH1, a neighboring gene that encodes for a transcription factor that relates strongly with carcinogenesis [31]. However, the regulation of gene expression through lncRNA transcript function is only one side of the coin [17]. Recent studies indicate that the genomic locus and/or nascent transcription of certain lncRNAs exert significant functions primarily through chromatin conformation and epigenetic regulations. In this context, NALT1 is also considered to exert locus-dependent functions on top of the functions that are mediated from its mature transcript [30]. As discussed below, the RNA-independent functions of lncRNAs reveal a largely unexplored regulatory layer that can further elucidate the complex mechanisms through which lncRNAs regulate their targets across various pathological contexts, offering the opportunity to discover novel and personalized therapeutic approaches.

### 2.1. The Long Non-Coding DNA Locus as the Key Player in Complex Diseases

#### 2.1.1. Regulatory Effects of lncRNA Loci in Cancer

Over the past few years, inspiring studies have proposed that ncRNA loci can upregulate or silence target genes through chromatin loops and/or embedded enhancer-like elements or insulators (Table 1). A representative example of complex lncRNA-mediated regulation that relates to carcinogenesis is CCAT1-L, which regulates the well-known oncogene MYC (myelocytomatosis oncogene), an oncoprotein that is overexpressed in various cancers [32]. In fact, in 2014 it was shown that this lncRNA interacts with the insulator protein CTCF (CCCTC-binding factor [33]), mediating the formation of a chromatin loop [34]. This loop includes the MYC locus and facilitates its interaction with neighboring enhancers. Surprisingly, the same research team also discovered the presence of a super-enhancer inside the CCAT1-L locus.

A series of experiments unraveled the mechanism by which CCAT1-L acts, demonstrating that its transcript participates in the extrusion of a chromatin loop between a super-enhancer embedded in the CCAT1-L and the promoter of MYC. This interaction results in the upregulation of the latter, fueling neoplastic transformation. The CCAT1-L case inspired the notion that not only the mature lncRNA transcripts but also their underlying DNA loci can exert independent regulatory functions. Soon afterward, it was revealed that several lncRNA loci act independently of their transcript either through embedded DNA elements or by their nascent transcription [43,44].

Another characteristic example refers to the murine lncRNA LockD, which is located 4 Kb downstream of its target gene, Cdkn1b. Cdkn1b is a protein-coding gene that encodes for p27, a key cell cycle regulator that is linked to carcinogenesis [45,46]. A series of in vitro experiments were able to provide mechanistic insights, deciphering the role of the LockD DNA locus, its nascent transcription, and the transcript itself for inducing the expression of Cdkn1b [35].

More specifically, a bovine growth hormone polyadenylation signal (BVG-polyA) was inserted downstream of the LockD’s transcriptional start site (TSS). This strategy, commonly abbreviated as PAS (polyadenylation signal), is routinely used to induce a premature stop in the transcription of a locus and consequently to decrease mature transcript levels [47]. In the case of LockD, insertion of a PAS indeed eliminated lncRNA transcript levels; however, the expression of the Cdkn1b mRNA was not affected, suggesting that neither the act of transcription nor the LockD transcript itself regulates Cdkn1b expression. Subsequently, the possibility of embedded regulatory elements at the LockD locus was put to test through next-generation Capture C, a sensitive technique that detects chromosomal contacts such as promoter–promoter interactions [48]. The results showed that the 5′ region of LockD spatially interacts with the promoter of the Cdkn1b, indicating that the underlying DNA sequence of the LockD 5′ region is sufficient for inducing Cdkn1b expression [35].

One of the most popular examples of a lncRNA locus that exerts significant functions on top of its transcript refers to PVT1. Like CCAT1-L discussed above, the plasmacytoma variant translocation 1 (PVT1) lncRNA also regulates MYC. Similar to its target mRNA, PVT1 is also an oncogene [49], since the transcript and its locus share distinct functions in cancer [50]. The PVT1 transcript participates in cell cycle regulation [51], associates with pancreatic neoplasms [28], and is overall considered a suitable biomarker for cancer [52,53,54]. Therefore, the role of the MYC-PVT1 regulatory axis is crucial, considering that many therapeutic strategies against neoplasia focus on MYC [55].

Recently, the regulatory effect of the PVT1 locus in MYC regulation was uncovered [36]. One of the main experiments relied on the use of CRISPRi (clustered regularly interspaced short palindromic repeats inhibition, [56]) in the PVT1 promoter region. This CRISPRi-PVT1 strategy employed a dCas9 endonuclease [57] combined with a KRAB domain [58] resulting in heterochromatinization and therefore silencing of the PVT1 locus [36]. Inhibition of PVT1 transcription was associated with a 40% increase in the proliferation rate of breast cancer xenografts, with MYC being responsible for accelerating cancer cell growth, as confirmed through ASO (antisense oligonucleotides [59]) and shRNAs (short hairpin RNAs [60]) knock-down (KD) strategies coupled with transcriptome analysis. Importantly, when the researchers blocked the transcript-dependent function of PVT1, no effect was observed on MYC regulation, proving the dispensability of the lncRNA itself or the act of its transcription in the underlying mechanism. Subsequently, a second CRISPRi experiment using only dCas9 was conducted to reduce the PVT1’s transcription, while leaving the DNA sequence and the chromatin state intact. That CRISPRi experiment did not alter MYC’s expression [36], suggesting that the open chromatin state of the PVT1 locus is important for downregulating the expression of MYC.

More focused functional experiments revealed an RNA-independent regulatory layer centered around a competition between the PVT1 and the MYC promoter for accessing common enhancer sequences (Figure 1A). More specifically, four enhancer elements are embedded downstream of the MYC gene, the most important of which is named 822E [36]. When the PVT1 promoter is not active, MYC is upregulated. Interestingly CRISPRa-mediated activation of the PVT1 promoter [57] results in MYC downregulation. Importantly, perturbation of the MYC promoter results in the downregulation of PVT1 [61]. Such a negative feedback loop is the result of a 3D chromatin conformation, which allows only one promoter to interact with the enhancer element, as confirmed with Hi-ChIP (*in situ* Hi-C followed by chromatin immunoprecipitation [62]) and 4C-Seq (chromosome conformation capture combined with high-throughput sequencing [63]). The bromodomain protein BRD4, which is also known to induce MYC’s expression, seems to act as a key molecule for the establishment of this chromatin loop [64]. Inhibition of the PVT1 promoter resulted in increased BRD4 occupancy at the MYC promoter, highlighting the importance of BRD4 for regulating MYC expression [36]. Furthermore, several mutations at the PVT1 promoter region (obtained by published cancer human genomes), collectively highlight the importance of an intact PVT1 locus for MYC upregulation and cancer progression [36]. Apart from their role in cancer, it is worth mentioning that the PVT1–MYC interactions were also found to participate in diabetes [65].

As presented in the case of PVT1, a typical mechanism through which a lncRNA locus regulates a downstream target is through the formation of chromatin loops. However, no such loop compares to the one orchestrated by the MYNRL15 lncRNA locus. This lncRNA was found to be expressed in six human leukemic cell lines. Its DNA locus embeds two candidate *cis*-regulatory elements (cCREs), named cCRE C1 and C2 [37]. Leukemic cell lines with and without perturbation of the cCRE C1 were analyzed with next–generation capture C. Although most of the chromatin interactions within a radius of 500 Kb were similar in wild type (WT) and edited cells, two novel and ultra-long-range interactions located 12 Mb and 15 Mb upstream of the MYNRL15 locus were observed only in edited cells (Figure 1B, [37]). These unique ultra-long-range interactions could be mediated by CTCF, given its occupancy at the MYNRL15 locus.

Identification of the distal target of MYNRL15 was performed through a combination of chromatin conformation, leukemia dependency information [37], and transcriptome analysis. This combined strategy highlighted four candidate target genes: IMP3, WDR61, COMMD4, and SNUPN, all of which were found to be downregulated upon C1’s perturbation. Importantly, LNA-GapmeRs (locked nucleic acid [66]) and shRNA KD approaches proved that the MYNRL15 transcript is not implicated in the observed downregulation of these target genes. However, CRISPR-mediated excision of the whole MYNRL15 locus from four leukemic cell lines, elicited an anti-leukemic effect, proving the importance of this DNA region in cancer. The same anti-leukemic phenotype was also observed after CRISPRi-mediated downregulation of MYNRL15 in patient-derived xenografts (PDXs, patient cancer cells implanted in mice [67]), which impaired the propagation of two PDXs in recipient mice, confirming the regulatory competence of this DNA locus across a number of leukemic cells independently of its corresponding lncRNA transcript [37]. Although the long-range interaction of MYNRL15 and its anti-leukemic effects are established, additional experiments are needed to fully explain the genome-wide chromatin conformation like Hi-C (high-throughput chromosome conformation capture technique [68]) and fluorescence *in situ* hybridization (FISH).

As already implied by the case of PVT1, certain lncRNA loci drive major regulatory functions, while their associated transcripts are the mere “by-products” of the transcriptional process lacking an established functional role. Interestingly, other studies support the existence of lncRNAs that regulate downstream targets through the combined function of the transcript and its DNA locus in a non-overlapping manner. Such an example refers to HASTER, an evolutionarily conserved DNA region, embedded in the HNF1A locus, which controls the transcription of HNF1A-antisense lncRNAs (collectively known as HASTER RNAs) both in mice and humans. These lncRNAs act *in trans* to regulate cell proliferation and have been associated with a variety of malignancies [69]. HNF1A (hepatic nuclear factor 1A) is a transcription factor that regulates liver-specific gene expression. Mutations causing dysfunction of this factor have been linked to colorectal cancer and resistance to therapy [70]. Thus, it is important to investigate the ways by which the HASTER locus affects HNF1A expression.

Beyond the RNA-dependent function of the HASTER lncRNAs, a more recent study revealed that the HASTER DNA locus itself is necessary for the independent downregulation of HNF1A in murine and human hepatic cells. The deletion of the main HASTER promoter both in vitro (human hepatic cell lines) and in mice resulted in a significant elevation of HNF1A mRNA and protein levels. The transcripts as well as the act of their transcription are dispensable, as confirmed both by GapmeRs (antisense oligonucleotides used to degrade the nuclear lncRNA transcripts [71]) and CRISPRi. The results from both experimental approaches did not reveal any effects on HNF1A expression levels, ruling out a regulatory role through the transcription of HASTER or its resulting transcript. Interestingly, HNF1A also binds to the HASTER promoter, inducing its activation and consequently establishing a negative feedback loop (Figure 2A). In this way, cells ensure precise control over HNF1A transcription, which itself encodes for an important pioneer-like transcription factor [39].

Collectively, these observations suggest that the DNA locus of HASTER plays a fundamental role in the regulation of HNF1A expression in hepatic cells.

This hypothesis is further supported by comparing the epigenetic marks between WT and HASTER knockout (KO) mice. In WT mice, the deposition of H3K4me3 associated with active promoters [72] was enriched in the HASTER and HNF1A promoters. Interestingly, in the HASTER-KO background, this epigenetic mark spreads to a downstream intronic enhancer (referred to as E element) and to an upstream CTCF-bindind element (referred to as C element), suggesting that the excision of the HASTER locus could facilitate the interaction of both elements with the HNF1A promoter. Indeed, UMI-4C (unique molecular identifier 4C) analysis, a technique that is used to capture the chromatin conformation [73], revealed increased interactions between the E element and the region upstream of HNF1A upon HASTER deletion, further illustrating an insulator-like function of the HASTER locus. Overall, the DNA locus can encompass either enhancers or insulators that in turn control the expression of target genes regulating important cell decisions independently of the lncRNA transcript.

The locus of the murine lncRNA Rroid (RNA demarcated regulatory region of Id2) refers to another example of how lncRNA loci orchestrate cancer-related processes [38]. Rroid lncRNA (annotated as Ak083360) was identified in innate lymphoid cells (ILCs), a lymphocyte class crucial for initiating a protective immune response against viruses, parasites, and bacteria [74]. Rroid is strongly expressed in ILCs (especially in group 1 ILCs) and NK (natural killers [75]). Its locus is located 220 Kb upstream of the target gene Id2, which in turn encodes for a DNA inhibitor of T and B cells’ expression [76]. It is known that transcription of the Rroid locus is regulated by interleukin 15 (IL-15), a cytokine specifically required for the homeostasis of group 1 ILCs [77]. A CRISPR Cas9 approach was used to engineer Rroid^−/−^ mice which, contrary to WT, were not responsive to IL-15 administration, suggesting that the Rroid locus partially mediates the response to IL-15 [38]. In addition, Rroid^−/−^ mice were associated with a vast decrease in the frequency and absolute numbers of NK cells in the spleen, liver, and lung of mice. Analysis of these cells revealed an accumulation of immature CD27^+^ NK cells along with a reduction in CD11b^+^ mature NK cells in these tissues.

At the molecular level, direct involvement of the Rroid transcript in the regulation of Id2 was disproved through LNA KD experiments in cultured NK cells, since no notable change was observed in Id2 expression, despite the significant reduction in the Rroid transcript levels. On the contrary, the observed immune cell changes were associated with epigenetic modifications both at the Rroid locus and the Id2 target promoter. More specifically, ChIP-qPCR for histone modifications at the Id2 promoter of NK cells revealed a significant decrease in H3K27ac in the promoter and H3K36me3 across the gene body in Rroid^−/−^ mice. ATAC-seq (assay for transposase-accessible chromatin using sequencing [78]) further confirmed the heterochromatic state of the Id2 promoter in homozygous murine mutants, which in turn was associated with a reduction in transcription factor occupancy at the Id2 promoter. Collectively, these experiments support the vital role of Rroid in the upregulation of Id2, which in turn contributes towards proper immune system regulation [79].

Interestingly, ATAC-seq also revealed the existence of two DNA regulatory elements (RE) that are embedded inside the Rroid locus. Homozygous deletion of RE1 in mice was not associated with an effect on Rroid’s transcript levels but led to a decrease in NK maturation and Id2 expression. In addition, a combination of 3C (chromatin conformation capture [80]), ChIP-Seq, and ChIP-qPCR (all representing techniques for investigating chromatin composition and conformation) proved that RE1 formed a long-range interaction with the Id2 promoter upon IL-15 stimulation. That interaction was STAT5 (signal transducer and activator of transcription 5, [81])-mediated, stating that in the presence of IL-15 signaling cues, STAT5 interacts with the Id2 promoter, provided that an accessible RE1 element is present (Figure 2B) [38]. Taken together, disruption of the RE1 element in the Rroid locus is highly associated with carcinogenesis and dysfunction of NK cells.

#### 2.1.2. Regulatory Effects of lncRNA Loci in Other Complex Diseases

Similar DNA-dependent regulatory mechanisms have also been described for lncRNAs involved in other complex diseases. The murine lncRNA Bendr (also referred to as linc1536 locus, [40]) stands for Bend4 regulation not dependent on RNA, and its target gene is Bend4, which in humans has been found to be related to infection-induced acute encephalopathy [82]. PAS insertion in the first intron of the Bendr locus not only eliminated spliced lncRNA levels but strongly reduced the levels of the Bendr lncRNA. However, no changes at the levels of its mRNA target were observed. In sharp contrast, the deletion of a 750 bp DNA stretch located within the lncRNA locus that includes the Bendr promoter, resulted in a more than 50% decrease in the Bend4 target levels. In accordance with these results, it was concluded that the promoter and its proximal region, rather than the spliced lncRNA transcript or the act of its transcription, are responsible for regulating Bend4 expression. In conclusion, Bendr represents an unrelated to cancer example that highlights the prevalent role that lncRNA loci can impose on the regulation of target genes, emphasizing again the transcript-independent function of ncRNAs.

### 2.2. Nascent lncRNA Transcription as a Regulatory Mechanism

As the multifaceted regulatory roles of lncRNAs are thoroughly explored, additional lines of evidence support the notion that apart from the underlying sequence, nascent transcription (the process of productively transcribing a DNA locus), may play a role in orchestrating target genes’ expression. Below, we review typical examples of lncRNAs for which the process of transcription exerts regulatory function independently of the transcript product (summarized in Table 2).

#### 2.2.1. Regulatory Effects of Nascent lncRNA Transcription in Cancer

The protocadherin α (Pcdhα) locus is a bona fide example of how nascent lncRNA transcription is significant for the regulation of critical neurophysiological functions. The Pcdhα cluster is important in neurons because its genes are stochastically expressed to promote a phenomenon known as self-avoidance [83], according to which neurons utilize different combinations of surface-exposed Pcdh proteins to distinguish between self and non-self neurites. All genes of the cluster contain two CTCF binding sites (CBSs), one proximal to the promoter (pCBS) and one embedded in the first exon (eCBS). An enhancer called HS5-1 (hypersensitivity site 5-1, [89]) resides downstream of the cluster and contains—two additional CBSs. Interestingly, the first exon of each Pcdhα gene also contains an extra promoter responsible for initiating the transcription of an antisense lncRNA [83]. In fact, as determined in vitro through CRISPR-activation (dCas9-VPR [90]), it is the antisense-lncRNA that triggers the expression of the sense-coding mRNA, suggesting the regulatory role that these lncRNAs may exert on their sense-coding counterparts.

To gain insights into the exact mechanism by which antisense lncRNAs regulate sense transcription in the cluster, the dCas9-VPR strategy was further expanded, to compare chromatin modifications and conformational changes upon sense or antisense promoter activation. In this context, ChIP-seq showed that when the antisense promoter was activated, CTCF occupancy increased, suggesting the possibility of CTCF-cohesin-mediated loop extrusion. Using methylated DNA immunoprecipitation (meDIP, [91]) to examine DNA methylation patterns in Pcdhα cluster, it was also observed that upon transcriptional activation of the antisense lncRNA, both CBSs of its respective alternate gene were significantly demethylated, allowing binding of CTCF. What is more, cHi-C (a modified Hi-C technique used for short loci analysis, [92]) revealed extended interactions between the exact, each time, alternate gene and the HS5-1 enhancer, when the antisense promoter was activated, further confirming the hypothesis of long-range chromatin looping facilitating enhancer–promoter interactions (Figure 3). Cohesin-mediated loop formation was also examined in vivo, as Rad21 (a cohesin domain)-depleted mice showed no loop formation at the Pcdhα locus. Thus, nascent transcription is essential for Pcdhα stochastic expression and normal neurites identification and it is worth noting that aberrant methylation of the locus has been reported in different types of cancer [93], such as in Wilms tumors [94].

The regulatory functions that stem from nascent lncRNA transcription are also reflected in the mechanism that controls X chromosome inactivation (XCI) [95]. The associated X-inactive-specific lncRNA transcript (XIST) is vital for XCI initiation during early development through its accumulation at the X chromosome [96]. Xist was recently found to be upregulated by the lncRNA five prime to Xist (Ftx) [84]. Although the Ftx transcript regulates many genes and is tightly associated with cancer [97,98,99,100], its DNA locus is important in XCI [84]. XCI is strongly correlated with the Ftx-XIST axis and its aberrant performance can lead to cancer, since downregulation of XIST results in decreased XCI and increased risk of autoimmune diseases and oncogenesis [101]. The significant role of XCI is further supported by the observation that in the absence of Ftx’s expression, female mice exhibit a phenotype that resembles human microphthalmia, a disease that is also linked to the X chromosome [102].

To thoroughly unveil the underlying regulatory mechanism, a CRISPR/Cas9 approach targeting the Ftx promoter in mouse ESC (mESC) was used [84]. FACS analysis (separation of the cells by flow cytometry [103]) revealed an increase of up to 30% in cell death in the double mutant (Ftx^−/−^) in comparison to the WT (Ftx^+/+^) cells. On the fourth developmental day, the Ftx^+/+^ cells were able to produce Xist RNA (XIST accumulation in the nuclei was 32%) but only 4% of Ftx^−/−^ cells experienced XIST accumulation in the nucleus. That was also validated by RNA-FISH experiments, as the mature Xist transcript was significantly downregulated in the absence of Ftx. Of note, LNA-Gamper experiments discarded the possibility that the Ftx lncRNA transcript participates in XIST’s regulation, yet the act of its transcription was found to be crucial [84]. The researchers conducted CRISPRi on mESC that effectively ceased transcription of the Ftx lncRNA without compromising the integrity of the genomic region. The transcriptional inhibition of Ftx led to a similar phenotype with the Ftx^−/−^ cells (decreased XIST accumulation), highlighting the importance of Ftx’s nascent transcription for expressing and accumulating XIST. Furthermore, transcription of Ftx in undifferentiated female mESC was associated with the formation of a 1 Kb chromatin loop that connects the promoters of Ftx and XIST, inducing XIST expression (Figure 4A). It should be noted, however, that currently, the chromatin conformation pattern upon inhibition of Ftx’s transcription is not assessed. In summary, even though the exact regulatory mechanism is not yet fully uncovered, Ftx lncRNA transcription upregulates the expression of XIST through the formation of a DNA loop, having a major impact on developmental processes [84,102].

Another interesting case that highlights the multifaceted role of ncRNA loci on various developmental processes refers to the antisense Igf2r RNA non-coding (Airn) locus, which is involved in the genomic imprinting of the insulin-like growth factor 2 receptor (Igf2r) gene in mice [85,104]. Under physiological conditions, Igf2r participates in the organogenesis of the cardiovascular [105,106,107] and the central nervous system [108], but it is also involved in carcinogenesis [109]. Interestingly, activation of Airn prohibits Igf2r’s expression. More specifically, the paternal allele of the Airn promoter is functional and results in the downregulation of Igf2r [85]. Since the Igf2r and Airn overlap and their promoters have opposite orientations in mice, binding of the RNA Pol II to the Airn promoter induces transcriptional interference of the Igf2r gene through steric hindrance (a phenomenon that here refers to transcriptional inhibition due to over-accumulation of RNA Pol II molecules) [85]. On the contrary, the maternal allele of Airn hosts an imprinted control element (ICE) in its promoter region [110], which epigenetically silences the promoter of Airn through methylation of CpG islands, thus allowing the activation of the Igf2r gene (Figure 4B).

The functional role of Airn’s productive transcription was further established through a PAS insertion strategy. When PAS was inserted between Airn’s and Igf2r’s promoters, causing Airn’s transcriptional termination before the Igf2r promoter, no effect was observed in the methylation of Igf2r. Hence, the Airn transcript is dispensable for the regulation of Igf2r, although it is known to directly regulate other target genes such as Scl22a3 and Prr18/Qk/Pde10a [111,112]. However, insertion of PAS downstream of the Igf2r promoter in the Airn locus terminated Airn transcription downstream of the Igf2r promoter, causing paternal imprinting of Igf2r. These results suggest that nascent transcription of the Airn is significant only when it permeates the Igf2r promoter. To fully address the details of that mechanism, the researchers inverted Airn’s promoter to match the orientation of Igf2r. In that way, Airn’s transcription does not intervene with Igf2r’s promoter and consequently, no effect was observed on the expression of Igf2r. Therefore, when Airn transcription surpasses the promoter of Igf2r, this leads to its imprinting on the paternal chromosome [85]. Since transcription accumulates RNA Pol II molecules on the Airn locus, it initiates steric hindrance at the entire Airn locus, inhibiting other RNA pol IIs from binding to the Airn embedded promoter of the Igf2r gene (Figure 4B [85]). Collectively, the Airn lncRNA transcript and its locus regulate distinct genes via separate mechanisms, featuring once again the complexity of the lncRNA-mediated regulation.

#### 2.2.2. Nascent lncRNA Transcription in Other Diseases

With an increasing number of lncRNAs that function via nascent transcription being identified, it is obvious that the extent of the transcriptional elongation can be crucial for initiating downstream effects. A characteristic example is the lncRNA Meteor. Meteor stands for mesendoderm transcription enhancer organizing region, whereas its target gene Eomes is a T-box transcription factor essential for establishing a functional cardiac mesoderm [113,114]. Eomes was recently found to participate in the long-term retainment of pathogenic CD4+ T cells in inflammation. Consequently, it was worth investigating the mechanisms by which the lncRNA Meteor regulates this factor. In this context, it was shown that minimal transcriptional initiation of the Meteor locus promotes the expression of Eomes in mESCs [86]. More specifically, when the promoter of Meteor is removed or inverted, Eomes levels are significantly decreased. Replacement of the endogenous Meteor promoter with a constitutive one (e.g., PGK), or alternatively engineering of KD models using dCas9, PAS alone, or PAS coupled with Rz (self-cleaving ribozyme, [115]), had no effect on Eomes levels, indicating that the presence of a functional promoter is sufficient for minimal initiation of Meteor’s transcription and normal Eomes expression (Figure 5A). Indeed, 4C analysis revealed that the absence or inversion of Meteor’s promoter reduces the spatial proximity between the Meteor and the Eomes loci, as minimal nascent transcription cannot be initiated. Collectively, these data indicate that minimal transcriptional initiation of the Meteor locus ensures accessibility to its promoter, which in turn regulates Eomes expression through chromatin looping. It should be noted, however, that the involvement of the Meteor transcript in the aforementioned Meteor–Eomes axis was not directly assessed and therefore cannot be excluded.

Beyond Meteor, the productive transcriptional elongation into the gene body of the murine lncRNA Maenli is required for the regulation of a downstream target gene [87]. Maenli stands for master activator of engrailed-1 in the limb and is transcribed from a conserved locus that resides about 250 Kb downstream of engrailed-1 (En1), a transcription factor that controls dorsal–ventral patterning in the limb [87]. Interestingly, deletion of the Maenli locus was identified in three patients suffering from limb malformations. Yet, surprisingly, these patients had the same phenotype as patients with a specific biallelic mutation in En1, supporting the notion that Maenli also regulates En1 also in humans.

To experimentally validate the dispensability of the DNA locus for the regulation of En1, the PAS sequence was inserted immediately downstream of Maenli’s TSS in mice. The results showed a 90% reduction in both Maenli and En1 levels, along with a double dorsal limb phenotype that was observed, indicating that Maenli’s nascent transcription or the transcript itself regulates En1’s expression. However, inverted PAS or exon and intron deletions at the Maenli locus had no effect on En1 expression. Surprisingly, when a GFP sequence was inserted prior to the PAS signal, spacing it further downstream from the TSS, a 70% reduction in En1 levels and a milder phenotype were observed (Figure 5B). These data support the notion that nascent transcription and, more specifically, its increased elongation, regulate En1 independently of the nucleotide sequence or the mature transcript. What is more, a comparison of WT and PAS knock-in (KI) mice, using ATAC-seq and ChIP-Seq, revealed significant differences in the deposition of active epigenetic marks at the En1 and Maenli loci, further confirming the regulatory role of Maenli’s transcription in the activation of En1. Inspired by the ThymoD mechanism (discussed in detail below), the research team utilized DamID seq (DNA adenine methyltransferase identification sequencing), a versatile tool for genome-wide DNA-protein interactions profile [116], to examine the possibility that the En1 locus is repositioned from the lamina to the nuclear interior, but no repositioning was observed. Nonetheless, both Meteor and Maenli represent two characteristic examples of lncRNAs that utilize nascent transcription to regulate downstream target genes both in mice and humans.

A different example of regulatory nascent lncRNA transcription refers to the bivalent locus upregulated by the splicing and transcription of an RNA (Blustr) lncRNA, or linc1319, which positively regulates the Scm-like with four Mbt domains 2 (Sfmbt2) gene [40]. Sfmbt2 encodes for a suppressor that is related to cancer progression [117,118]. The transcript sequence of Blustr does not participate in the regulation of Sfmbt2 as indicated through a series of KD experiments, which did not affect the expression of the latter. However, the length of the transcription is important. Insertion of a PAS sequence close to the Blustr promoter significantly downregulated Sfmbt2. In addition, the deletion of a splicing site at the first intron resulted in a 92% reduction in the mature Blustr transcript and an 85% reduction in Sfmbt2, while the deletion of splicing sites further downstream had no effect. Therefore, the essential part of the Blustr lncRNA sequence is located on the splicing site of the first intron, which is proposed to be important for accumulating transcription factors at the Blustr gene [40].

### 2.3. Examples of lncRNA That Combine Diverse Mechanisms to Collectively Regulate Target Genes in Cancer

Growing evidence highlights the existence of lncRNAs that exert their RNA-independent functions through a combination of embedded DNA elements in their sequence along with nascent transcription of their locus (summarized in Table 3). In such cases, the transcriptional process first promotes chromatin modifications, enabling respective chromatin conformational changes along with specific interactions between regulatory elements.

ThymoD is a complex, yet fascinating example of a murine lncRNA for which both nascent transcription and embedded regulatory elements play a significant role in T-cell differentiation. This lncRNA locus resides 850 Kb downstream of Bcel11b (B-Cell Lymphoma/Leukemia 11b), a gene encoding a transcription factor that controls early T-cell development [123]. Furthermore, Bcl11b serves as a tumor suppressor in T-cell acute lymphoblastic leukemia [124]. Therefore, it is important to elucidate all possible mechanisms of its regulation. Interestingly, the ThymoD locus encompasses an enhancer of Bcl11b [125]. To investigate the mechanism through which ThymoD regulates Bcl11b, mice with a PAS-KI inside the lncRNA locus were engineered. Disruption of ThymoD’s transcription decreased Bcl11b mRNA levels, suggesting that the act of transcription and/or the mature transcript are necessary for normal Bcl11b expression [119]. LNAs designed to target and eliminate the mature ThymoD transcripts confirmed the dispensability of this transcript in the regulation of Bcl11b [126].

Independent experiments unraveled a model according to which nascent transcription initially promotes the demethylation of specific CpG islands in the ThymoD locus, allowing CTCF to bind and recruit cohesin. As a result, a chromatin loop embracing both the Bcl11b and its enhancer is extruded. Notably, this loop repositions the Bcl11b enhancer from the lamina (heterochromatin) to the nuclear interior (euchromatin) facilitating its interaction with the Bcl11b promoter (Figure 6A), as shown with Hi-C and principal component analysis (PCA, a dimensionality reduction tool [127]).

These conclusions were also supported by the study of homozygous mice for PAS insertion at the ThymoD locus, which readily developed lymphomas and leukemias, indicating that the act of ThymoD transcription but not its transcribed nucleotide sequence, is responsible for enhancing the accessibility of this region, facilitating activation of Bcl11b by the ThymoD-embedded enhancer through conformational changes in the chromatin. Taken together, the ThymoD example clearly supports the crucial role of lncRNA nascent transcription in orchestrating chromatin conformation changes irrespectively of its produced transcript.

A different lncRNA that functions through a combination of a DNA element and the act of transcription is lincRNA-p21 [120]. One of the genes that this lincRNA regulates is p21 (or Cdkn1a), a TF that inhibits cell cycle progression. Although the p21 protein has an anti-proliferative role in WT cells, its expression is dysregulated in several cancer types [128]. Therefore, its lncRNA-mediated regulation was further analyzed. It was shown by CRISPR experiments that p53 regulates (i) lincRNA-p21, through binding to a proximal p53 response element (abbreviated as P53RE), and (ii) p21 through binding to the proximal as well as a distal P53RE [120]. Of note, the proximal P53RE of the p21 gene corresponds to an enhancer-like element [129], while the lincRNA-p21 and the p21 loci interact via chromatin loops as shown with 3C analysis.

Targeted experiments were performed to reveal the exact details of the underlying mechanism. In mice, PAS insertion by CRISPR at the lincRNA-p21 locus resulted in the downregulation of lincRNA-p21, yet with no observable effect on p21 mRNA. Subsequently, the TWISTER ribozyme strategy [130] was also used for the degradation of the transcript of lincRNA-p21, again with no effect on p21 expression. Although the mature lincRNA-p21 transcript itself does not participate in the underlying mechanism, its nascent transcription is vital for the regulation of p21. CRISPRi (using the dead RNAs strategy [131]) was associated with a decrease in p21 expression, indicating that nascent transcription is in part responsible for p21 regulation. Interestingly, several embedded elements in the lincRNA locus have been identified including a CEBPB (CCAAT enhancer binding protein beta) element and a MyoD [132] binding site, which have been implied to participate in the p21–lincRNA-p21 axis [129].

Maintaining the same context, the presence of embedded elements at the lincRNA locus was examined. A series of P53RE deletions within the lincRNA as well as in other downstream conserved sequences resulted in the downregulation of p21 expression, highlighting the importance of p53 and supporting the existence of multiple regulatory elements at the lincRNA-p21 locus [120]. It should be noted, however, that the mutation of a conserved sequence at the first exon of lincRNA-p21 also reduced p21 levels, suggesting the existence of additional regulatory sequences in this region [120]. Importantly, according to the Hi-Cap [133] and Hi-C results, when lincRNA-p21 and p21 are expressed, their loci physically interact via an intra-chromatin looping [129]. Taken together, the nascent transcription of lincRNA-p21 along with P53RE elements coordinate the regulation of p21’s expression upon activation by p53 (Figure 6B) [120,129,134]. Once initiated, the lincRNA-p21-mediated downregulation of the p21 protein results in faster cell proliferation, an observation that agrees with the inhibitory role of p21 in cancer [120].

Another unusual locus refers to the RUNXOR lncRNA. RUNXOR is an unspliced 260 Kb long transcript with several DNA elements in its locus, which regulates the runt-related transcription factor 1 (RUNX1) gene [121]. RUNX1 is dysregulated in acute myeloid leukemia [135] and has a regulatory role in different subtypes of breast cancer, functioning both as a tumor suppressor and as an oncogene [136,137]. The RUNX1 locus has two promoters, P1 and P2, which control the expression of three isoforms (a, b, and c) and are embedded within the RUNXOR locus. The P1 promoter is close to RUNXOR’s TSS while P2 is located approximately 50 Kb downstream. To add to the complexity of this region, two main enhancers (named RE1 and RE2) and a silencer also operate inside the RUNX1 locus (Figure 7A).

Because of the extreme length of RUNXOR, no plasmid activation system can be applied to achieve its overactivation [121]. Thus, CRISPR was used to insert the cytomegalovirus (CMV) sequence [138] in the promoter of RUNXOR in a breast cancer cell line. Cis stimulation of RUNXOR resulted in P1 activation fueling its upregulation, but no change was observed in the status of the P2 promoter. Next, the chromatin interactions between the P1 and 2 promoters with the embedded regulatory elements were investigated upon RUNXOR’s overactivation. According to 3C analysis, when RUNXOR is overexpressed, local remodeling of chromatin architecture is initiated. Yet the P1 promoter was found to interact only with the RE2 element, while the P1–RE1 interaction was decreased [121].

RUNXOR overexpression in breast cancer cells also stimulates the demethylation of CpG islands embedded at P1, along with a 20-fold increase in H3K4me3, indicating that P1 (but not P2) epigenetic activation is responsible for the observed upregulation of the RUNX1c isoform. It should be noted that participation of the mature RUNXOR transcript in the regulation of RUNX1 cannot be ruled out, as it was found to interact with the RUNX1 locus via its 3′-UTR, although the exact mechanism through which this interaction participates in the RUNXOR-RUNX1 regulatory axis is unclear [139]. In conclusion, the act of transcription of the RUNXOR lncRNA serves as a molecular switch for the transcription of specific RUNX1 isoforms, building up to the regulatory complexity of this locus.

It is known that cells often utilize a “feedback loop strategy” to achieve precise control of gene expression. This phenomenon has also been observed for lncRNAs. A characteristic example is the modulation of the HOXA gene cluster by the lncRNA Haunt (HOXA upstream non-coding transcript) in mice. HOXA genes are induced by the retinoic acid (RA) morphogen and their induction is essential in the formation of the anteroposterior body axis in vertebrates [140]. Dysregulation of HOXA genes relates to numerous malignancies such as blood cancer [141]. Therefore, identification of all possible regulators of the cluster is fundamental for discovering novel therapeutic opportunities. A recent study showed that the Haunt locus contains enhancers that interact with HOXA loci, promoting their expression upon RA signaling [122]. Yet, when the Haunt lncRNA is expressed, its transcripts accumulate in cis to prevent enhancer–promoter interactions that in turn govern HOXA expression, establishing an antagonistic negative feedback loop that spatiotemporally regulates the HOXA cluster (Figure 7B).

To unravel the exact mechanism of HOXA regulation, a series of in vitro experiments were conducted. CRISPR-Cas9-mediated excisions of DNA fragments with various lengths revealed that when small fragments of the Haunt locus were deleted in the presence of RA, Haunt levels were reduced but HOXA expression remained unaffected. In contrast, deletion of larger DNA fragments parallel to RA signaling, resulted in a significant decline of HOXA levels as determined by RNA-seq, pointing towards the existence of embedded DNA elements at the lncRNA locus. Interestingly, a cDNA KI approach that was used to exogenously increase Haunt transcript levels, failed to rescue HOXA levels, further supporting the prevalent regulatory role of the embedded DNA elements. On the other hand, when a CAG-KI strategy (homologous recombination of a CMV constitutive promoter) was used to overexpress Haunt [142]) replacing its endogenous promoter, HOXA gene expression was reduced indicating that the mature Haunt transcript negatively regulates HOXA genes. These contrasting effects were explained by 3C analysis, which revealed that in WT ESCs the Haunt and HOXA loci are in close proximity and their interaction slightly increases upon RA signaling. Disruption of Haunt transcription through the PAS KI strategy leads to a dramatic increase in the same chromatin interactions whereas in the CAG KI model, the opposite effect is observed, illustrating once more the opposing roles that the Haunt DNA locus and its mature transcript exert. Finally, chromatin isolation RNA purification (ChIRP, [143]) revealed the direct binding of the endogenous Haunt transcript to HOXA loci, shedding light on the mechanism through which it attenuates HOXA gene expression. Taken together, these data clearly state that in some cases, the DNA locus of a lncRNA antagonizes its mature transcript, establishing a precise control mechanism for vital developmental processes.

### 2.4. The Regulation of Heart Formation through the Contrasting Effects of Two lncRNA Loci on a Common Target Gene

In cellular contexts beyond carcinogenesis, two neighboring lncRNA loci can also exert opposing functions in the regulation of a common target gene. Such is the case of HAND2 (heart and neural crest derivative 2), which encodes for a transcription factor with a crucial role in heart development [144]. Abnormal expression of HAND2 due to loss-of-function mutation leads to familial dilated cardiomyopathy [145]. Because of the crucial role that HAND2 exerts, the precise regulation of its expression is ensured through several genes including two lncRNAs, located upstream (Upperhand, Uph) and downstream (Handsdown) of its locus. Interestingly, both lncRNAs regulate HAND2 expression in an RNA-independent manner utilizing some of the previously mentioned mechanisms (Figure 8A).

More specifically, the DNA locus of Uph, encompasses a variety of embedded DNA elements including a super-enhancer, a branchial arch enhancer, and other cardiac-specific enhancer sequences [146]. Transcription of this locus is controlled by a bidirectional promoter that transcribes Uph antiparallel to the HAND2 mRNA. Nascent transcription of Uph was shown to be essential for HAND2 regulation, based on data derived from a PAS KI mouse model that was engineered [41]. A more recent study, however, presented results from three different mice KO models showing that the Uph DNA locus, rather than its nascent transcription or the resulting transcript, is mainly responsible for upregulating HAND2 expression in a precise and spatially controlled manner [42]. Despite the observed disagreement regarding the involvement of its transcript, both studies support a model through which altered Uph activation leads to an aberrant expression of HAND2 and subsequently to severe heart malformation and lethality (Figure 8B), highlighting once more the critical role that a lncRNA locus may facilitate, with or without its transcript, in the regulation of downstream target genes.

Although HAND2 expression is positively regulated by the Upperhand locus, a different lncRNA named Handsdown or Hnd that is located downstream of HAND2, has also been shown to participate in the regulation of HAND2. Shortly after the induction of HAND2 expression, cells rely on the Hnd lncRNA locus for its downregulation [88]. The importance of the whole Hnd locus was indicated via its excision through CRISPR-Cas9 (D10A) nickase technique, in mice embryos [147]. Heterozygous KO of Hnd resulted in hyperplasia in the right ventricular wall, but homozygous embryonic KO was associated with lethality. Moreover, the Hnd transcript is dispensable, since both LNA GapmeRS, which reduced its levels by 90%, or exogenous overexpression had no observable effect on the expression of HAND2 [88]. Yet, transcriptional activation of the endogenous locus through dCas9-VP64 [148], or insertion of a pA-MAZ sequence (short 99 bp transcriptional stop signal) immediately downstream of the first exon of Hnd, affected HAND2 levels, collectively suggesting that transcription of Hnd is essential for regulating HAND2 expression. Additionally, 4C experiments demonstrated that only when the Hnd gene is active does its locus interact both with the HAND2 locus and surprisingly also with the 5′ region of Uph [149]. It should be noted that according to a different study, opposing results were presented concerning the phenotype of Hnd locus perturbation, as neither heart hyperplasia nor lethality was observed [150]. These conflicting results could be attributed to the differences in the size and area of the deletions that took place. In total, both research teams excluded a role for the Hnd transcript in the regulatory mechanism, yet more studies are needed to fully clarify the link between the Hnd lncRNA locus and its nascent transcription in HAND2 regulation [88,150]. Collectively, HAND2 is a particularly significant heart-specific transcription factor that must be precisely expressed to ensure proper cell differentiation for a normally formed heart. Thus, the up- and downregulation of HAND2 by Uph and Hnd lncRNAs is time-dependent and relies on the different cell types that are ordinarily generated (Figure 8C).

## 3. Conclusions

The case studies presented above, suggest that the lncRNA transcript is not always exclusively responsible for mediating ncRNA function in complex diseases such as cancer. This conclusion is particularly important for the biomedical scientific community, considering that a common challenge of ncRNA biology is to decipher the function of differentially expressed lncRNAs in pathological specimens. This is not a straightforward ordeal even when careful experimental designs are in place, since frequently mild or no phenotypes are observed upon inhibition of relevant lncRNA transcripts despite their strong association with clinical manifestations in patient cohorts. In this respect, we propose that embedded DNA elements in the lncRNA loci, independently or in coordination with the respective transcripts, can participate in lncRNA-mediated function, uncovering novel ways through which the scientific community can weaponize RNA biology against complex diseases. The opportunities for utilizing complex lncRNA mechanisms in biomedical research are further highlighted by the fact that i) current RNA therapeutic schemes focus entirely on transcript inhibition and ii) certain lncRNA loci demonstrate distinct or overlapping functions in the regulation of their downstream target genes compared to the regulatory role of their respective transcript. Therefore, diverging roles of the lncRNA transcripts and their DNA loci can be observed for the same target gene depending on the cellular context.

On the other hand, lncRNA transcripts and their loci can regulate components of the same pathway but can also regulate independent genes with differentroles in pathology. Thus, although the study of lncRNA function currently focuses on the characterization of their transcript, it is of equal interest to widen our experimental strategies with the inclusion of approaches like LNAs, PAS insertion, and CRISPR to assess the regulatory role of their associated loci. This is particularly imperative considering that lncRNAs contribute to pathology also through the accumulation of a significant mutational burden in their loci, since they are not evolutionary constrained by the genetic code [151]. For instance, the PVT1 locus frequently hosts mutations in patients with various forms of cancer, adding a layer of genetic predisposition in the regulatory role of lncRNA loci in complex disease [151]. Future progress on lncRNA mutations under the prism of ncDNA loci function in disease, could prove particularly fruitful in delivering novel diagnostic applications or genetic markers, allowing the scientific community to complement and significantly expand the currently existing arsenal of genetic predisposition of complex diseases.

## Figures and Tables

**Figure 1 cancers-16-02728-f001:**
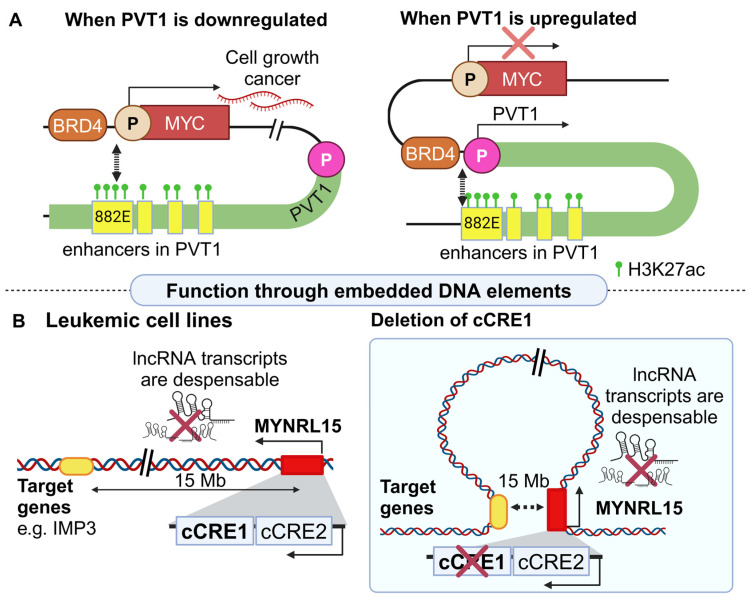
lncRNA-mediated regulation through embedded DNA elements (**A**). The PVT1 locus hosts enhancers like 822E. When PVT1 transcription is inactive, the MYC promoter gains access to the enhancers and its mRNA is produced (left). Upon PVT1’s activation, chromatin is reorganized (BRD4), ensuring that the enhancers are not in close proximity to the MYC promoter, leading to its inactivation (right) (**B**). MYNRL15 has two DNA elements in its locus (cCRE1 and cCRE2), while its distal target genes are located 12–15 Mb downstream of its locus (left). Deletion of cCRE1 facilitates the formation of an ultra-long-range loop spanning 15 Mb, allowing the interaction of MYNRL15 with its downstream distal target genes (right) (created with Biorender.com on 27 July 2024).

**Figure 2 cancers-16-02728-f002:**
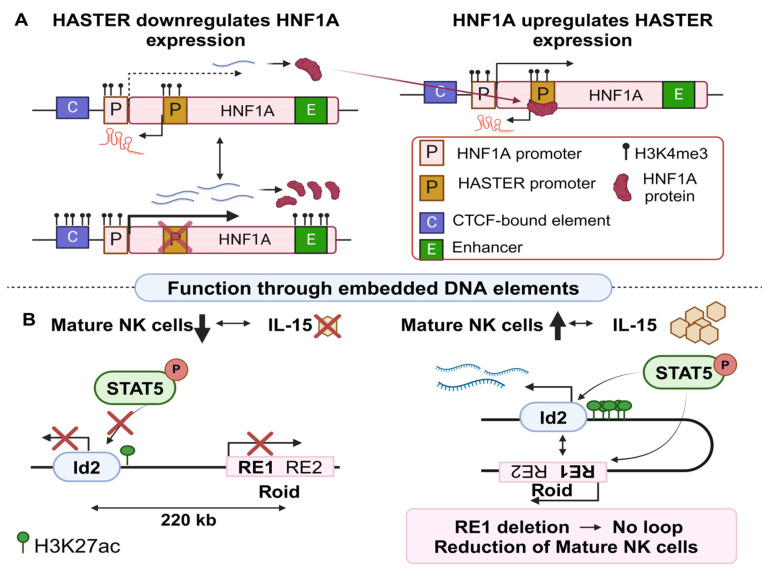
LncRNA regulation through embedded DNA elements (**A**). HASTER lncRNA locus is embedded in the HNF1A gene. HASTER promoter negatively regulates the expression of HNF1A (upper left). Once produced, HNF1A upregulates HASTER through direct binding to its promoter (right), establishing a negative feedback loop. When the promoter of HASTER is deleted (lower left) HNF1A expression is significantly upregulated. (**B**). The loss of mature NKs is linked to IL-15 deficiency. The phosphorylated (p)STAT5 cannot bind to the Id2 promoter, therefore the Id2–Rroid lncRNA does not interact and Id2 is suppressed (left). In the presence of IL-15, pSTAT5 binds to Id2 and Rroid (right), bringing together the Rroid’s RE1 DNA element to the promoter of Id2, causing upregulation of the latter in mature NK cells (created with Biorender.com on 27 July 2024).

**Figure 3 cancers-16-02728-f003:**
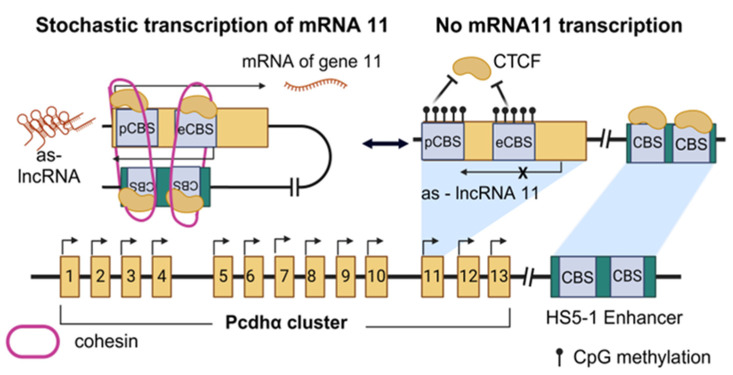
lncRNAs regulate target genes through nascent transcription. This example refers to stochastic expression of the Pcdhα gene 11. Transcription of an antisense lncRNA (as-lncRNA) (left) gives rise to a CTCF-mediated loop formation that brings together the downstream HS5-1 enhancer and the Pcdhα gene 11, allowing the expression of the latter. However, when the as-lncRNA is not transcribed (right), no loop extrusion is observed (created with Biorender.com on 27 July 2024).

**Figure 4 cancers-16-02728-f004:**
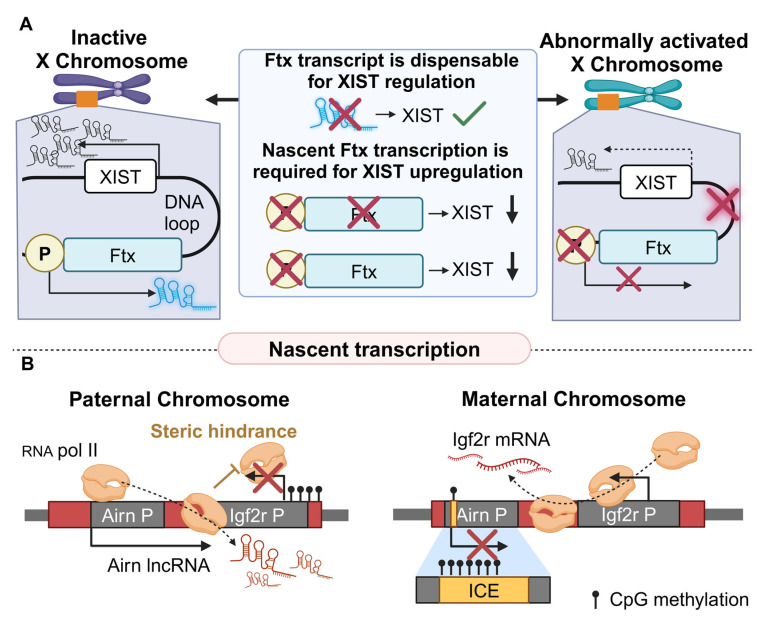
Examples of lncRNAs that regulate target genes related to cancer through nascent transcription (**A**). Ftx transcription results in a loop formation with the XIST locus leading towards the upregulation of the latter. However, inhibition of Ftx’s transcription results in downregulation of XIST. Therefore, XIST cannot accumulate properly and XCI is not normally performed. (**B**). In the paternal chromosome (left) promoter is accessible and the Airn lncRNA is transcribed. Due to the proximity of Airn and Igf2r promoters, Igf2r cannot be transcribed (RNA pol II-mediated steric hindrance phenomenon). In the maternal chromosome (right) Airn promoter is epigenetically inactivated by the ICE region and Igf2r is normally transcribed (created with Biorender.com on 27 July 2024).

**Figure 5 cancers-16-02728-f005:**
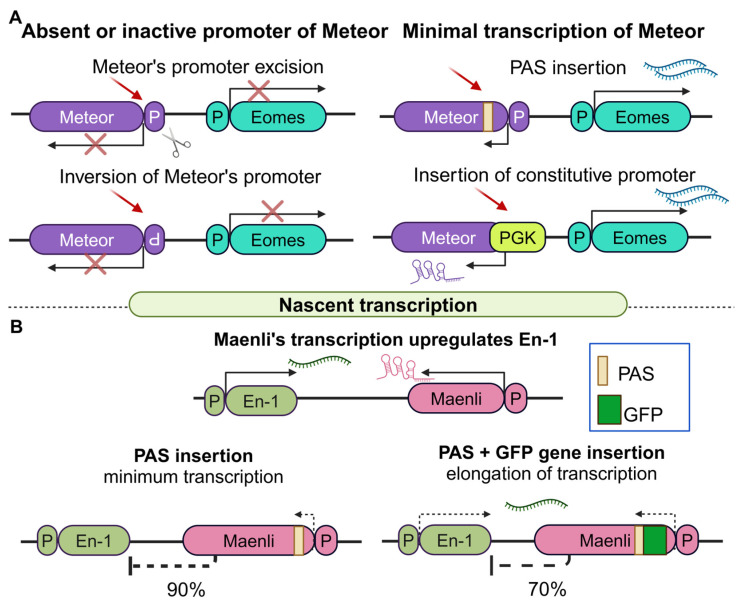
Examples of lncRNAs that regulate target genes related to diseases through nascent transcription (**A**). Minimal transcription of the lncRNA Meteor is necessary for proper expression of its target gene, Eomes. When the promoter of Meteor is deleted or inverted, Meteor is not transcribed and as a result, Eomes is not expressed (left). On the contrary, when minimal transcription of Meteor takes place (upon PAS insertion), or when its promoter is replaced by a constitutive one (PGK) Eomes is expressed (right). (**B**) Transcriptional elongation at the gene body of Maenli upregulates its target gene En-1. PAS insertion (left) prohibits transcriptional elongation leading to a 90% decrease in En-1 expression. However, when PAS is incorporated further downstream due to insertion of a GFP sequence gene (right), transcriptional elongation goes on a little further resulting in a milder decrease in En-1 levels (70%) (created with Biorender.com on 27 July 2024).

**Figure 6 cancers-16-02728-f006:**
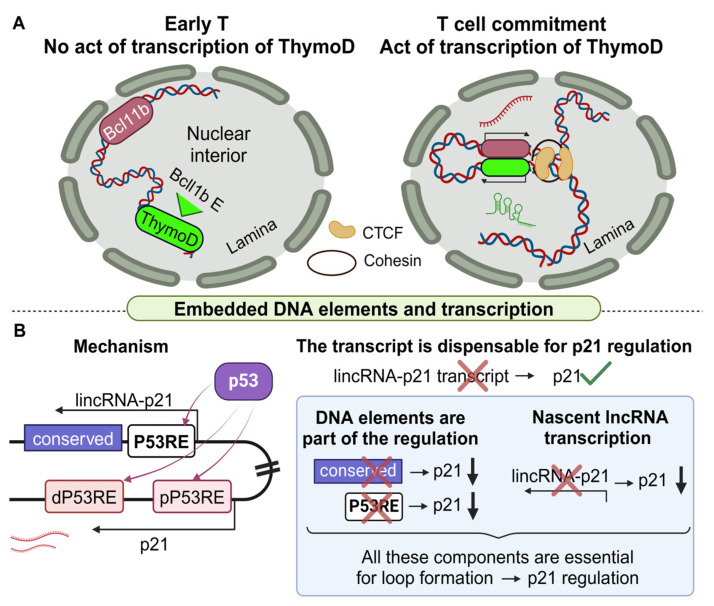
lncRNAs that regulate target genes through a combination of embedded DNA elements and nascent transcription. (**A**). When the ThymoD lncRNA is not transcribed (left) its locus and the embedded Bcl11b enhancer localizes in the nuclear lamina. Nascent transcription of ThymoD (right) initiates a CTCF-mediated loop, bringing together Bcl11b and its enhancer. This interaction leads to Bcl11b upregulation and T-cell commitment. (**B**). LincRNA-p21 upregulates p21 upon p53 signaling through the formation of a loop. The existence of conserved DNA sequences with embedded p53 response elements (P53REs) along with nascent lincRNA-p21 transcription are all indispensable for the loop formation, yet the lncRNA transcript has no regulatory role (created with Biorender.com on 27 July 2024).

**Figure 7 cancers-16-02728-f007:**
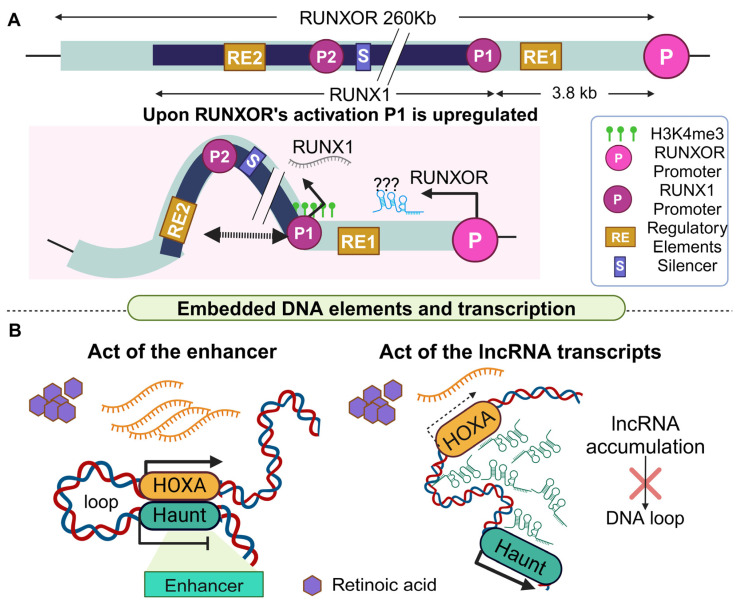
RUNXOR and Haunt regulate target genes through a combination of strategies. (**A**). The architecture of the 260 Kb long RUNXOR DNA locus is complex, consisting of the RUNX1 gene, its two promoters (P1 and P2), two regulatory elements (RE1 and 2), and a silencer (S). RUNXOR’s transcription associates with a chromatin conformational change, allowing acetylation and interaction of P1 with RE2, and consequently expression of RUNX1 from P1. (**B**). The Haunt lncRNA locus encompasses an enhancer that upregulates HOXA genes. When Haunt is not transcribed (left) a chromatin loop is formed, bringing Haunt in close proximity to the HOXA cluster, allowing the expression of the latter. However, when Haunt is transcribed (right), Haunt transcripts accumulate and bind directly to the chromatin of the region, leading to the dissociation of the loop (created with Biorender.com on 27 July 2024).

**Figure 8 cancers-16-02728-f008:**
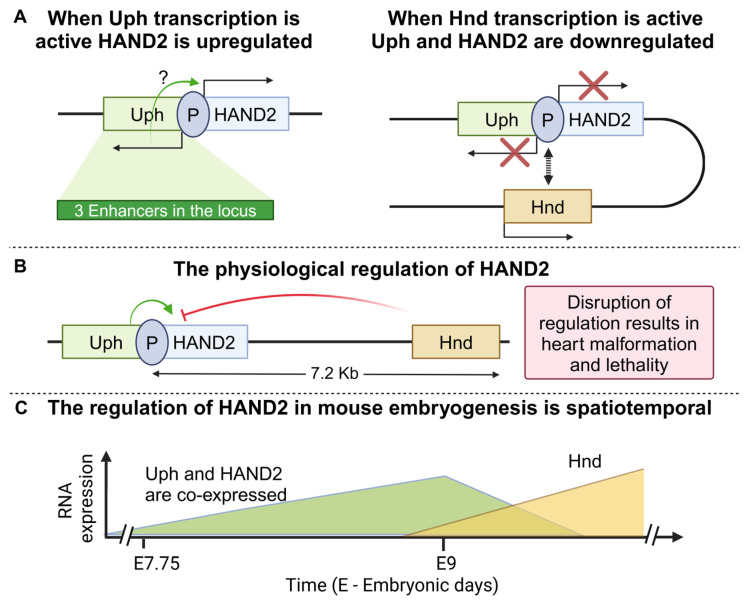
HAND2 is regulated by two lncRNAs loci (**A**). Induction of Upperhand (Uph) results in HAND2 upregulation and coexpression of both genes (left). When Handsdown (Hnd) is transcribed a DNA loop is formed resulting in HAND2/Uph downregulation (right) (**B**). The physiological regulation of the Uph/HAND2/Hnd locus (**C**)**.** The diagram is an illustration of the spatiotemporal expression of Uph over Hnd. Uph is primarily expressed at E7.75 in the cardiac crescent and then its levels decrease when Hnd begins to express at E9 in crest cells (created with Biorender.com on 27 July 2024).

**Table 1 cancers-16-02728-t001:** The DNA element as the main mechanism of function.

lncRNA (L)	Target Gene (TG)	Distance **	Effect on TG	Mechanism of Function	Physiological Processes	Dysregulated Phenotype of TG	Ref.
**LockD**	Cdkn1b (p27)	+4 Kb	Down	Promoter-like element, chromatin interaction	Cell cycle progression	Cancer	[35]
**PVT1**	MYC	+58 Kb	Down	Chromatin conformation, promoter competition for enhancer	Cell proliferation	Cancer	[36]
**MYNRL15**	IMP3 WDR61 COMMD4 SNUPN	+12–15 Mb	Down	CTCF-mediated long-range looping (12–15 Mb radius)	RNA splicing, DNA replication	Anti- leukemic effect	[37]
**Rroid**	Id2	−220 Kb	Up	Epigenetic modifications, long distance interactions	NK cell lineage	Cancer	[38]
**HASTER**	HNF1A	+2 Kb *	Down	Epigenetic modifications, insulator like element, stability of HNF1A	Hepatic cell differentiation	Cancer	[39]
**Bendr**	Bend4	ND, + TSS	Up	Enhancer-like element in the Bendr promoter/5’ proximal region	Promotion of germ cell differentiation	Acute encephalopathy	[40]
**Upperhand** **(Uph, Hand2os1)**	HAND2	−123 bp (antisense)	Up	Super-enhancer—promoter interactions	Heart development	Heart morphogenesis leading to lethality	[41,42]

** Distance of L from TG, − upstream, + downstream, ND = not determined, * = estimated distance TSS to TSS, Ref. = references.

**Table 2 cancers-16-02728-t002:** The act of transcription as the main mechanism of function.

lncRNA (L)	Target Gene (TG)	Distance **	Effect on TG Expression	Mechanism of Function	Physiological Processes	Dysregulated Phenotype of TG	Ref.
**Pcdhα**	Pcdha cluster	Antisense overlapping	Up	Nascent lncRNA transcription, target demethylation, CTCF – cohesin looping,	Cell surface identity signals of neurons	Possible cancer association Wilms tumor	[83]
**Ftx**	Xist	+195 Kb *	Up	Loop formation, nascent transcription	X chromosome inactivation (XCI)	Cancer	[84]
**Airn**	Igf2r	−30 Kb	Down	Perturbation of RNA pol II from the target gene promoter	Genomic imprinting, development	Cancer	[85]
**Meteor**	Eomes	+70 Kb	Up	Minimal transcriptional initiation	Cardiac mesoderm and neuronal differentiation programs	CD4+ T cell retainment in inflamation	[86]
**Maenli**	En1	−251 Kb	Up	Transcription elongation, epigenetic modifications	Dorsal–ventral patterning in the limb	Brain abnormalities and limb malformation	[87]
**Blustr**	Sfmbt2	+5 Kb	Up	Transcription elongation, epigenetic modifications and 5’ splicing site of intron 1	ND	ND	[40]
**Handsdown**	HAND2	+7.2 Kb	Down	CTCF independent chromatin interactions (yet not fully understood)	Heart development	Heart hyperplasia, lethality	[88]

** Distance of L from TG, − upstream, + downstream, ND = not determined, * = estimated distance TSS to TSS, ECS embryonic stem cell, mECS mouse embryonic stem cell, Ref. = references.

**Table 3 cancers-16-02728-t003:** lncRNA locus regulatory effects through combinations of mechanisms of function.

lncRNA (L)	Target Gene (TG)	Distance of L from TG	Regulation of the TG	Mechanism of Function	Physiological Processes	Dysregulated Phenotype	Ref.
**ThymoD**	Bcl11b	+850 Kb *	Up	Demethylation, CTCF binding, cohesin, loop, chromatin reposition	T-lineage determination	Lymphoma and leukemia	[119]
**lincRNA p21**	Cdk1a (p21)	+12 Kb	Up	Nascent lncRNA transcription, p53 elements	Apoptosis, cell cycle control, hypoxia	Increased cell proliferation	[120]
**RUNXOR**	RUNX1	−3.8 Kb	Up (RUNX1c)	Enhancer and promoter interaction in the target gene	Cell development, hematopoiesis	Cancer (breast cancer and leukemia)	[121]
**Haunt**	HOXA cluster	−40 Kb	DNA locus: Up lncRNA transcript: down	Nascent transcription, enhancer promoter interaction	Development	Cancer	[122]

− Upstream, + downstream, ND = not determined, ESC embryonic stem cell, Ref. = references. * = estimated distance TSS to TSS.

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
