# Peer review of "RNA-Independent Regulatory Functions of lncRNA in Complex Disease"

_cancers, 2024, doi:10.3390/cancers16152728_

Round 1

Reviewer 1 Report

Comments and Suggestions for Authors

In the review presented, the authors seek to thoroughly examine and illustrate, through a multitude of examples, the ways in which the DNA locus and/or the nascent transcription of a long non-coding RNA (lncRNA) can play crucial or even dominant roles in the regulation of downstream target genes in mammals. This exploration aims to shed light on the significant impact these mechanisms have on gene regulation.  

Unfortunately, the manuscript requires extensive revision because it can be difficult to follow, both in terms of writing style and the organization of the paragraphs, whose content often does not reflect their titles. For example, why is chapter 2 divided into "2.1.1. Cancer and cancer related lncRNAs" and "2.1.2. Immunology and cancer"... what do these two titles mean? And why in the third paragraph of chapter 2, titled "2.1.3. LncRNAs in other diseases," there are additional examples of lncRNAs still related to cancer? What example from paragraph 2.2.2, "Nascent lncRNA transcription in Sex determination and imprinting of genes," justifies the part of the title that mentions sex determination? Moreover The introductory part of chapter 2 is actually an example like those presented in the following paragraphs A representative example of complex lncRNA-mediated regulation that relates to 89 carcinogenesis is CCAT1-L ….)

Additionally, there are sentences that don't make any sense to me, such as: "Embedded DNA elements in the lncRNA loci and the act of transcription, independently or in coordination, can participate in the function of lncRNAs. Cis elements and/or  transcription itself do not participate in the function of the lncRNA... I would say they are two separate things...

Comments on the Quality of English Language

I'm not a native speaker, but I still believe that the writing needs improvement, particularly in terms of fluency and clarity of many sentences.

Author Response

Reviewer #1

In the review presented, the authors seek to thoroughly examine and illustrate, through a multitude of examples, the ways in which the DNA locus and/or the nascent transcription of a long non-coding RNA (lncRNA) can play crucial or even dominant roles in the regulation of downstream target genes in mammals. This exploration aims to shed light on the significant impact these mechanisms have on gene regulation.

Major remarks and point-to-point-response

Reviewer point 1 Unfortunately, the manuscript requires extensive revision because it can be difficult to follow, both in terms of writing style and the organization of the paragraphs, whose content often does not reflect their titles. For example, why is chapter 2 divided into "2.1.1. Cancer and cancer related lncRNAs" and "2.1.2. Immunology and cancer"... what do these two titles mean? And why in the third paragraph of chapter 2, titled "2.1.3. LncRNAs in other diseases," there are additional examples of lncRNAs still related to cancer? What example from paragraph 2.2.2, "Nascent lncRNA transcription in Sex determination and imprinting of genes," justifies the part of the title that mentions sex determination? Moreover The introductory part of chapter 2 is actually an example like those presented in the following paragraphs ( A representative example of complex lncRNA-mediated regulation that relates to 89 carcinogenesis is CCAT1-L ….)

Author response: We fully appreciate the effort of the reviewer and his/hers comments that significantly improve the clarity of our work. We also apologize for structuring our original manuscript in such a complex way. We have simplified the structure and number of our sections in the revised form of our manuscript, ensuring that in their current form they encapsulate the content in a streamlined and more uniform manner. The reviewer can find the new section titles in the revised lines 205, 225-226, 438, 454, 465, 580, 656 and 841-842. Moreover, we restructured our text, merging the CCAT1-L story with the remaining cases of the 2.1.1 section (Regulatory effects of lncRNA loci in cancer, revised lines 230-245) separating it from the introductory text of section 2 (revised lines 207-224)

Reviewer point 2 Additionally, there are sentences that don't make any sense to me, such as: "Embedded DNA elements in the lncRNA loci and the act of transcription, independently or in coordination, can participate in the function of lncRNAs. Cis elements and/or  transcription itself do not participate in the function of the lncRNA... I would say they are two separate things...

Author response: Well taken, we apologize for the confusion. We re-wrote this part of our text to improve its clarity (revised lines 910-914). We also consulted an English native speaker colleague to edit our revised manuscript. We hope that our editing efforts combined with the significantly increased number of illustrations in our revised manuscript will simplify and clarify our message to the reader. Thank you.

Reviewer 2 Report

Comments and Suggestions for Authors

This review focuses on the study titled " RNA-independent regulatory functions of lncRNA in complex disease". This review provides a concise summary of specific studies, categorized based on the extent to which the associated transcript is involved in regulating pathological or physiological processes. These studies collectively emphasize the significance of non-coding RNA in mediating functions that go beyond the RNA level.

Decision: Major revision

Comments:

1-     I suggest adding a paragraph to highlight the biogenesis and function of lncRNAs in the introduction section

2-     More Figures are needed to summarise all signaling pathways and elucidate the biogenesis and function of lncRNAs

3-     It is generally a good idea to provide details on the review search methodology in a systematic review or a review article at the end of the introduction. This information can help readers assess the rigor and comprehensiveness of the search process and evaluate the validity and generalizability of the review findings. Some key details that could be included in the review search methodology section are: 

a) Search period: The time frame during which the search was conducted, including the start and end dates of the search.

b) Types of reviews used: The types of reviews that were included in the search, such as systematic reviews, meta-analyses, narrative reviews, or scoping reviews.

c) Language: The language(s) of the reviews that were included in the search. This could include a description of any language restrictions that were used in the search process.

d) Databases used: The databases that were searched, such as PubMed, Embase, Cochrane Library, or Web of Science.

e) Applied keywords: The keywords and search terms that were used to identify relevant reviews. This could include a list of the specific search terms that were used in the search process.

4-     A list of abbreviations should be added to the manuscript

5-     Some important publications in this area of study are missed in this manuscript. For example:

Role of long non-coding RNAs in pancreatic cancer pathogenesis and treatment resistance-A review. https://doi.org/10.1016/j.prp.2023.154438

Long non-coding RNAs and rheumatoid arthritis: pathogenesis and clinical implications. https://doi.org/10.1016/j.prp.2023.154512

6-     There is no prospect of the effect of lncRNAs on diseases. What is the new research direction of lncRNAs? It is necessary to gain an insight into the research direction.

7-     To ensure the article is current and pioneering, it is recommended to update the literature cited. Keeping the references up-to-date will enhance the timeliness and relevance of the article's content.

Author Response

Reviewer #2

This review focuses on the study titled " RNA-independent regulatory functions of lncRNA in complex disease". This review provides a concise summary of specific studies, categorized based on the extent to which the associated transcript is involved in regulating pathological or physiological processes. These studies collectively emphasize the significance of non-coding RNA in mediating functions that go beyond the RNA level.

Decision: Major revision

Major remarks and point-to-point-response

Reviewer point 1      I suggest adding a paragraph to highlight the biogenesis and function of lncRNAs in the introduction section

Author response: We would like to sincerely thank the reviewer for his/her fruitful suggestions that collectively improve text flow and strengthen the message of our manuscript to the reader. We have added a new paragraph dedicated to lncRNA biogenesis and supported with relevant citations in lines 156-169 of our revised text. Thank you.

Reviewer point 2 More Figures are needed to summarise all signaling pathways and elucidate the biogenesis and function of lncRNAs

Author response: Well taken, we have significantly increased the number of our figures, through the creation of new content and reorganization of the previous panels. In the revised manuscript the reader will find in total eight figures compared to the original three, covering 87% (14 out of 16) of our selected lncRNA cases in the main text compared to the original 37%. We did not include lncRNAs with similar mechanisms to those already illustrated or cases with incompletely described mechanisms. We hope that these illustrations, along with the two tables, collectively improve the message of our manuscript towards the reader. Thank you. 

Reviewer point 3 It is generally a good idea to provide details on the review search methodology in a systematic review or a review article at the end of the introduction. This information can help readers assess the rigor and comprehensiveness of the search process and evaluate the validity and generalizability of the review findings. Some key details that could be included in the review search methodology section are: 

  1. a) Search period: The time frame during which the search was conducted, including the start and end dates of the search.
  2. b) Types of reviews used: The types of reviews that were included in the search, such as systematic reviews, meta-analyses, narrative reviews, or scoping reviews.
  3. c) Language: The language(s) of the reviews that were included in the search. This could include a description of any language restrictions that were used in the search process.
  4. d) Databases used: The databases that were searched, such as PubMed, Embase, Cochrane Library, or Web of Science.
  5. e) Applied keywords: The keywords and search terms that were used to identify relevant reviews. This could include a list of the specific search terms that were used in the search process.

Author response: Thank you for the useful comment indeed. We have revised our text, adding a new paragraph with all the requested information regarding our survey in the revised lines 193-203.

Reviewer point 4 A list of abbreviations should be added to the manuscript

Author response: We have incorporated a list of used abbreviations in revised lines 38-121.

Reviewer point 5 Some important publications in this area of study are missed in this manuscript. For example:

Role of long non-coding RNAs in pancreatic cancer pathogenesis and treatment resistance-A review‏. https://doi.org/10.1016/j.prp.2023.154438

Long non-coding RNAs and rheumatoid arthritis: pathogenesis and clinical implications‏. https://doi.org/10.1016/j.prp.2023.154512

Author response: We have added both manuscripts as citations [28] and [29] in our revised text.

Reviewer point 6 There is no prospect of the effect of lncRNAs on diseases. What is the new research direction of lncRNAs? It is necessary to gain an insight into the research direction.

Author response: Well taken, we have revised our main text, emphasizing on the clinical perspective of certain lncRNA (e.g revised lines 211-224, 271-274, 357-359, 436-437, 493-496, 510-515, 550-552 etc) along with the existing information in both tables. However, to strengthen this important point and to better emphasize on the research direction that we propose, we also modified our conclusion section (revised lines 904-935), providing tangible reasons for turning the RNA research of the biomed field towards the RNA-independent mechanisms of ncRNA loci. Importantly, we complement this information with the citation of two previous reviewing efforts from our lab (citations [17] and [151]), which together with the existing manuscript provide a complete outline regarding the role and operation of lncRNA transcripts, ncRNA loci and ncRNA mutations in the pathology of complex diseases. 

Reviewer point 7 To ensure the article is current and pioneering, it is recommended to update the literature cited. Keeping the references up-to-date will enhance the timeliness and relevance of the article's content.

Author response: We fully agree. In the revised form of our manuscript we cite 151 articles in total. 77% percent of them is published within the last five years and 30% of all articles is published within 2023 and 2024. The latter were not present in our original manuscript. These percentages are typical for similar reviews published in Cancers. Please note that we put special care in citing recent articles and experimental approaches however, we are limited by the publication year of our selected case studies since the examples of ncRNA loci with an established mechanism are rare in the literature. We hope that the reviewer will understand these limitations and we thank you for the careful consideration of our work.

Round 2

Reviewer 1 Report

Comments and Suggestions for Authors

It seems to me that the authors have addressed my comments and made changes to the manuscript. I think it can be accepted

Reviewer 2 Report

Comments and Suggestions for Authors

The authors have addressed all my concerns. I consider the manuscript acceptable for publication